# Molecular Detection and Genetic Characterization of Potential Zoonotic Swine Enteric Viruses in Northern China

**DOI:** 10.3390/pathogens11040417

**Published:** 2022-03-30

**Authors:** Gebremeskel Mamu Werid, Yassein M. Ibrahim, Hongyan Chen, Lizhi Fu, Yue Wang

**Affiliations:** 1State Key Laboratory of Veterinary Biotechnology, National Poultry Laboratory Animal Resource Center, Harbin Veterinary Research Institute, Chinese Academy of Agricultural Sciences, Harbin 150069, China; ashenafymamo@gmail.com (G.M.W.); yassin8322@gmail.com (Y.M.I.); chenhongyan@caas.cn (H.C.); 2Chongqing Academy of Animal Science, Chongqing 408599, China; 3College of Veterinary Medicine, Southwest University, Chongqing 400715, China

**Keywords:** astrovirus, encephalomyocarditis virus, kobuvirus, sapovirus, epidemiology, coinfection

## Abstract

Despite significant economic and public health implications, swine enteric viruses that do not manifest clinical symptoms are often overlooked, and data on their epidemiology and pathogenesis are still scarce. Here, an epidemiological study was carried out by using reverse transcription-polymerase chain reaction (RT-PCR) and sequence analysis in order to better understand the distribution and genetic diversity of porcine astrovirus (PAstV), porcine encephalomyocarditis virus (EMCV), porcine kobuvirus (PKV), and porcine sapovirus (PSaV) in healthy pigs reared under specific pathogen-free (SPF) or conventional farms. PKV was the most prevalent virus (51.1%, 247/483), followed by PAstV (35.4%, 171/483), then PSaV (18.4%, 89/483), and EMCV (8.7%, 42/483). Overall, at least one viral agent was detected in 300 out of 483 samples. Out of the 300 samples, 54.0% (162/300), 13.0% (39/300), or 1.0% (3/300) were found coinfected by two, three, or four viruses, respectively. To our knowledge, this is the first report of EMCV detection from porcine fecal samples in China. Phylogenetic analysis revealed genetically diverse strains of PAstV, PKV, and PSaV circulating in conventional and SPF farms. Detection of swine enteric viruses with a high coinfection rate in healthy pigs highlights the importance of continuous viral surveillance to minimize future economic and public health risks.

## 1. Introduction

Swine enteric viruses cause significant economic losses in the pork industry. Viruses that do not cause severe clinical signs in pigs may have public health and/or epidemiological implications [1,2]. PAstV, EMCV, PKV, and PSaV, for example, are non-enveloped single-stranded RNA viruses that cause gastroenteritis infection in pigs, with or without clinical signs [1,2,3]. In addition, PAstV, EMCV, PKV, and PSaV have been found to be genetically closely related to human viral strains and hence are suspected of zoonosis and/or thought to be potentially zoonotic [1,4,5]. Besides having higher strain diversity, high mutation rate, and higher recombination capacity of genomes, these viruses lack the fragile lipid envelope, giving them the ability to stay infectious for a long time in suitable environmental conditions.

PAstV is a non-enveloped virus, 28–30 nm in diameter, containing a single-stranded, positive-sense RNA genome of 6.4–7.3 kb in size [6]. Astroviruses infect a wide range of wild and domestic animals, including humans. PAstV belongs to the family *Astroviridae* and genus *Mamma Astrovirus*. With five genetic lineages (PAstV1-5), PAstVs are genetically diverse and suspected to possess zoonotic potential [7].

EMCV is a naked, single-stranded positive-sense RNA virus with a genome size of approximately 7.8 kb, classified under the *Picornaviridae* family and *Cardiovirus* genus [8,9]. EMCV causes myocarditis, encephalitis, and reproductive failure in a wide range of host species, including humans [10]. The successful replication of porcine ECMV in human cardiomyocytes [11], detection of antibodies against porcine EMCV in humans [12], and the isolation of human EMCV strains that are closely related to porcine EMCV [13] signal the zoonotic potential of EMCV.

PKV, also called *Aichivirus C*, is a member of the *Picornaviridae* family. PKV is a non-enveloped, single-stranded, positive-sense RNA virus with a genome size of approximately 8.2–8.3 kb [14]. PKV has been linked with gastroenteritis [15]. The available information on cross-species transmission and viral recombination events suggests that PKV has zoonotic potential [16,17].

PSaV is a non-enveloped, single-stranded, positive-sense RNA virus, with icosahedral capsid and a genome size of approximately 7.3–8.5 kb [18]. PSaV, classified under the *Caliciviridae* family and *Sapovirus* genus [8], is an emerging causative agent of gastroenteritis in pigs. Due to its high genetic diversity, recombination ability, and the presence of genetic and antigenic similarity between human and swine strains, PSaV is suspected to be zoonotic [19,20].

Despite previous reports on these viruses, the epidemiological characteristics of each virus in pigs and the coinfection rate have not been reported yet. Here, in order to understand the epidemiology and distribution of PAstV, EMCV, PSaV, and PKV in healthy pigs, a surveillance study was conducted by RT-PCR-based screening of fecal samples collected from SPF and conventional swine farms. In order to explore the genetic diversity of newly detected viruses, nucleotide sequences were analyzed and compared with previously reported strains.

## 2. Results

### 2.1. Prevalence of PAstV, EMCV, PKV, and PSaV in Pigs Raised on SPF and Conventional Farms

The presence of PAstV, PKV, EMCV, and PSaV in 483 fecal specimens collected from pigs reared in SPF and conventional farms were checked by RT-PCR, as described previously [21,22,23,24]. Before the large-scale screening of samples, randomly selected positive samples were confirmed by sequencing. At least one target virus was detected in 62% (300/483) of the tested samples, of which, 51.1% (247/483), 35.4% (171/483), 18.4% (89/483), and 8.7% (42/483) were positive for PKV, PAstV, PSaV, and EMCV, respectively (Table 1, Appendix A).

Compared to SPF pigs (*n* = 310), the pigs kept at conventional farms (*n* = 173) showed a higher prevalence for each of the target viruses (Table 1). PKV was found to be the predominant virus in both farm types; however, the prevalence of PKV was higher in conventional farms (72.8%, 126/173) compared to SPF farms (39%, 121/310). The pigs reared in conventional farms showed higher prevalence of PAstV (56.1%, 97/173), PSaV (29.5%, 51/173), and EMCV (19.1%, 33/173) compared to SPF pigs, which were 23.9% (74/310) for PAstV, 12.3% (38/310) for PSaV, and 2.9% (9/310) for EMCV (Table 1). A statistically significant difference (*p* < 0.05, *X*^2^ = 11.27) in the overall viral detection rate was found between the pigs reared under SPF (48.7%, 151/310) and conventional farms (86%, 149/173). Whereas, from the four viruses tested, only the prevalence of EMCV showed a statistically significant difference (*p* < 0.05, *X*^2^ = 13.714) between the pigs reared under SPF (2.9%, 9/310) and conventional farms (19%, 33/173).

### 2.2. Prevalence of PAstV, EMCV, PKV, and PSaV in Different Age Groups

PKV was found to be the most prevalent virus in all of the age groups, except in weaners, with a prevalence of 71.5% (113/158) in piglets, 45.6% (52/114) in fattening pigs, and 40% (20/50) in sows (Table 1). In contrast to the other age groups, weaners showed a higher prevalence of PAstV, EMCV, and PSaV, with rates of 45.3% (73/161), 10.6% (17/161), and 25.5% (41/161), respectively (Table 1). A statistically significant difference (*p <* 0.05) in the prevalence of PAstV, EMCV, PKV, and PSaV was observed among the different age groups.

### 2.3. Coinfection of PAstV, EMCV, PKV, and PSaV

Furthermore, the coinfection rates in pigs reared under conventional and SPF farms were analyzed. A coinfection rate of 68% (204/300) was observed from the pigs that tested positive for at least one target virus. While pigs coinfected with two or three viruses were detected in both farm types, pigs coinfected with all four of the viruses were detected exclusively in the conventional farms (Figure 1). A coinfection rate of 53.0% (80/151) was observed in the SPF pigs, which is significantly different (*p* < 0.05, *X*^2^ = 15.05) from the pigs kept under conventional farms (83.2%, 124/149). The coinfection rate was also found to be significantly different (*p* < 0.05, *X*^2^ = 55.6) among the age groups. A coinfection rate of 61.7% (71/115), 84.8% (84/99), 59.7% (37/62), and 50.0% (12/24) was observed in piglets, weaners, fattening pigs, and sows, respectively (Table 1). No sow was found to be coinfected with more than two types of viruses (Figure 1).

From a total of 300 pigs that were infected with at least one target virus, 30% (91/300) of them were infected with PKV and PAstV, 8% (24/300) with PKV and PSaV, 7% (21/300) with PAstV and PSaV, and 9% (26/300) with PAstV, PKV, and PSaV (Figure 2). Unlike PSaV, PKV, and PAstV, which were detected in both single and mixed infections, EMCV was detected exclusively in mixed infections (Figure 2).

### 2.4. Sequencing and Phylogenetic Analysis

Based on farm type, approximately 10–20 positive samples for PAstV, EMCV, PKV, and PSaV were sequenced. Nucleotide sequences of the detected strains and additional GenBank reference strains were aligned, and phylogenetic trees for each virus were generated using neighbor-joining and the Kimura 2-parameter model with bootstrap analysis (1000 replicates).

A phylogenetic tree for PAstV was built using a partial sequence of the ORF1b/ORF2 gene from 12 PAstV strains identified in this study (PAstro-1 to PAstro-12) and 46 reference strains. All of the identified PAstV strains were classified under the clade that reflects the PAstV4 genotype. The identified PAstV strains displayed a distinct clustering pattern within the PAstV4 genotype. PAstro-5 and PAstro-10 were found closely related to PAstV4 strains from China (KX060809) and shared more than 96% nucleotide identity. While PAstro-1, 2, 3, 4, 6, 11, and 12 were grouped with Chinese strains (MW962975 and MK613068) and had 96.8–97.8% nucleotide identities, PAstro-7 and PAstro-8 were found to be closely related with PAstV4 strains from India (MG003343), with 85.5–87.9% nucleotide identity. Similarly, PAstro-9, with 98.4% nucleotide identity, was found to be closely related with another PAstV4 strain from India (MG003344), suggesting the genetic and geographic diversity of PAstV4 genotypes (Figure 3A).

Based on the partial nucleotide sequence of the 3D gene, a phylogenetic tree for EMCV was constructed using 11 identified sequences (EMCV-1-11) and 30 reference sequences. All of the EMCV strains that were detected in this study clustered with strains from China, Japan, South Korea, Belgium, and the USA and shared 99.4–100% nucleotide identity (Figure 3B).

A phylogenetic tree for PKV was constructed based on the partial nucleotide sequence of the 3D region using 13 identified strains (PKV1-13) and 45 reference strains. Although the identified PKV strains share more than 90.2% nucleotide identity, they were clustered under two distinct clades. PKV1, PKV2, and PKV3 were clustered closely with strains from China (KM051987, NC_027054, KP260507) and shared 93.9–96.1% nucleotide identity. The other 10 remaining stains clustered distinctly in another clade (Figure 3C), indicating different genetic variants of PKV.

The phylogenetic tree for PSaV was constructed based on the capsid region using 20 identified strains (Sapo-1 to Sap-20) and 55 reference strains. The identified PSaV strains clustered into three distinct clades, indicating genotype-based clustering. Sapo-2, Sapo-3, Sapo-4, and Sapo-20 clustered under the same clade with strains from Japan (LC215881 and LC215884) and shared 73.1–91.4% nucleotide identity. Sapo-1 and Sapo-16 to Sapo-19 were also grouped under one clade with PSaV from Japan (LC215883) and shared 88.3–89.2% nucleotide identities. Whereas Sapo-5 to Sapo-15 clustered closely with strains from China (MW285642) and shared 96.2–96.4% nucleotide identities (Figure 3D).

## 3. Discussion

In the current study, 483 fecal samples were tested and the prevalence, coinfection rate, and genetic diversity of PAstV, EMCV, PKV, and PSaV have been explored.

Despite the lack of evidence of zoonosis, the available limited information suggests that PAstV is a potentially zoonotic emerging virus associated with enteritis [25] and encephalomyelitis [26,27] in pigs. This is because of the genetic similarity of PAstV and human astroviruses [28,29], the possibility of recombination between PAstV and human astroviruses [7,28], and the concurrent detection of human and animal astroviruses in sludge and wastewater [7], all of which indicate the possibility of pig–human transmission. PAstV affects pigs of all age groups and has been detected from different tissue types [30,31]. In the current study, an overall prevalence of 35.4% was found for PAstV. Similarly, an overall prevalence of 34.2% (67/196) has been reported from the Czech Republic [3]. However, previous studies indicate that PAstV has a divergent prevalence ranging from 6.5% (32/488) in Thailand [2] to 92.2% (383/411) in Slovakia [32]. In China alone, an overall prevalence of PAstV was 56.4% (300/532) in Guangxi province [33], 46.3% (101/218) in Hunan Province [34], 17.6% (133/757) in Sichuan province [35], and 10.7% (23/215) in Anhui province [36]. Similar to other countries, a divergent prevalence of PAstV has been reported from South Korea (19.4%, 25/129) [37], Canada (79%, 76/96) [38], Croatia (89%, 81/91) [39], Germany (20.8%, 25/120) [4], Italy (67.4%, 163/242) [1], USA (64%, 326/509) [29], and India (17.5%, 21/120) [40]. These indicate that, with varying detection rates, PAstV is prevalent globally. Some risk factors, such as age [41], coinfection [30], and farm type, could play a role in the prevalence of PAstV. In the current study, compared to SPF pigs (23.9%, 74/310), the pigs that were reared under conventional farms showed a higher PAstV prevalence (56.1%, 97/173). Such differences in PAstV detection rate between farm types might be attributed to the higher hygienic practices in SPF farms. Though there is no previous report on PAstV detection from SPF farms, this study indicates that farm type is one of the risk factors for PAstV infection. In a study of over 22 weeks duration, an age-dependent PAstV fecal shedding was observed [41]. Similarly, a statistically significant difference in the prevalence of PAstV was observed among the age groups tested in the current study. Whereas, in other reports from USA and China, no significant age-based difference of PAstV prevalence was found [29,33]. For a comprehensive understanding of the risk factors of PAstV infection, in addition to age and farm type, further studies that address the associated risk factors, such as season of the year, husbandry practices, host genetics, and their interaction, might be required.

The phylogenetic analysis of the ORF1b/ORF2 partial region of PAstV revealed the presence of genetic diversity among the detected viral strains. According to the reference sequence, all of the PAstV strains identified in this study belong to the PAstV4 genotype, highlighting the presence of multiple PAstV4 lineages in China. Similarly, Zhao et al. [42] and Su et al. [31] from China reported the presence of genetically divergent PAstV4 genotypes in diarrheic pigs. These findings imply that Chinese PAstV4 strains have undergone genetic changes and may now be the dominant strains circulating in both diarrheic and non-diarrheic pigs in China.

Though there is no evidence of EMCV zoonosis, exposure to mice and primates [43,44], proximity to pig farms [45], and animal-related occupations, such as swine veterinarians, zoo employees, and hunters [46,47], have all been linked to EMCV infection in humans. Moreover, the fact that human EMCV strains were found to be closely related to porcine EMCV strains [13] suggests the zoonotic potential of EMCV. Infection with EMCV causes myocarditis, encephalitis, sudden death, and reproductive disorder in pigs [48,49]. Considering its economic and potential public health importance, understanding the epidemiology of EMCV and the associated risk factors could play an important role in controlling the virus. Thus, in the current study, the overall prevalence of EMCV was found to be 8.7% (42/483), which is similar to the reports from Germany (4.2%, 5/120) [4] and China (11.6%, 11/95) [50]. Compared to SPF pigs (2.9%, 9/310), a significantly higher EMCV prevalence was found in the pigs reared in conventional farms (19.1%, 33/173). Similar to the other viruses, the lower prevalence of EMCV in SPF farms might be attributed to the improved farm management conditions. Though EMCV has already been reported previously in pigs with myocarditis and/or reproductive failure [51,52,53], this is the first report of EMCV detection in healthy pigs in China. Based on previous reports [51,54], EMCV strains in China were grouped into two lineages. According to the phylogenetic analyses, all of the EMCV strains detected here were clustered along with porcine EMCV strains from China, Japan, South Korea, Belgium, and the United States, which is consistent with previous reports [51,55,56]. Additionally, the detected EMCV strains were found to be closely related to the strains from dogs (China, KU664327), mice (China, KF836388 and USA, DQ288856) and chimpanzees (USA, KM269482), indicating lineage-based clustering. Experimentally, one EMCV strain infected mice, pigs, and macaques and showed similar pathology results [54]. EMCV has been shown to overcome the species barrier in zoo outbreaks involving multiple animal species [57,58,59]. Other studies also linked EMCV infection in rodents to EMCV outbreaks in pigs [60,61]. Similarly, the current lineage-based clustering patterns could suggest a wider virus–host range and the possibility of cross-species transmission.

The sequence homology between animal kobuviruses and human kobuviruses [62] and the evidence of the transmission of bovine kobuvirus to pigs [16,17] suggest interspecies transmission and/or the zoonotic potential of kobuviruses, including PKV. PKV, detected in both diarrheic [3,16] and non-diarrheic pigs [16,63], has been linked to gastroenteritis. In the current study, the overall prevalence of PKV was found to be 51.1%, which is consistent with previous reports from Hungary [63], Japan [16], Korea [64], and Italy [15], but lower than the previous reports from the Czech Republic [3] and Slovakia [14]. Compared to the current study, a lower prevalence of PKV was reported from China [65], Vietnam [66], Italy [67], and East Africa [68]. These variations in the prevalence of PKV might be attributed to the type of farm management, animal physiological status, and other risk factors. In this study, compared to pigs under SPF farms (39.03%, 121/310), a higher prevalence (72.8%, 126/173) of PKV was found in pigs that were kept under conventional farms. Among the different age groups, the prevalence of PKV was found to be higher in piglets (71.5%, 113/158), indicating an age-dependent PKV infection.

Phylogenetic analysis revealed that the detected PKV strains were classified into two distinct clades, which is consistent with previous studies [15,64,65]. The detected PKV variants PKV1, PKV2, and PKV3 were found to be closely related to Chinese PKV strains, indicating geographic-based clustering. The remaining PKV strains detected in this study, on the other hand, clustered uniquely, suggesting the presence of a distinct and possibly new PKV lineage.

The higher genetic diversity of PSaV compared to human sapoviruses [69], the coinfection of oysters by both PSaV and human sapoviruses [69], and evidence of recombination between PSaV and human sapoviruses [5,70] suggest the zoonotic potential of PSaV. PSaV is a genetically divergent swine enteric virus that is commonly identified in many porcine farms. The current study observed an overall PSaV prevalence of 18.4% (89/483). A similar prevalence of PSaV was reported in China (14.37%, 22/153) [71] and Europe (11.1%, 117/1050) [19]. A slightly lower prevalence of PSaV was reported in the Czech Republic (10.2%, 20/196) [3], China (9.0%, 36/402) [72], Slovakia (9%, 37/411), and Korea (6.5%, 37/567) [73]. Whereas a higher prevalence of PSaV was reported in Brazil (23.7%, 40/169) [74] and the USA (62%, 385/621) [75]. The differences in PSaV prevalence across different countries might be due to the different risk factors of PSaV infection. For instance, in the current study, the prevalence of PSaV was found to be higher in conventional farms (29.5%) compared to SPF farms (12.3%). Age was another important risk factor for PSaV infection, indicating the susceptibility of young pigs.

The phylogenetic analysis of the partial sequence of the capsid region of PSaV showed the presence of genetically divergent PSaV genogroups GIII circulating in China. PSaV-GIII genetic divergence and higher prevalence have previously been reported [76,77,78], which is consistent with the findings of this study. The detected PSaV strains were grouped into three distinct clades. The Chinese reference strain (MW285642) and some newly detected strains were clustered into a single clade, suggesting location-based clustering. On the other hand, the remaining PSaV strains that were detected in this study were found to be clustered with Japanese reference strains (LC215883 and LC215884), implying genotype-based clustering.

In both SPF and conventional swine farm management conditions, mixed infection by swine enteric viruses was found to be common in healthy pigs. In contrast to PSaV, PKV, and PAstV, EMCV was only detected mixed with other viruses, suggesting mixed infection may play a role in EMCV replication, but this remains to be confirmed. While no sows were found to be infected with more than two viruses, a higher rate of coinfection was found in weaner pigs. The higher coinfection rate in weaner pigs might be attributed to the absence of maternal immunity after weaning [79]. Except for EMCV, no association was found between the farm type and virus prevalence. In contrast, age and farm type were significantly associated with coinfection, indicating that risk factors change during mixed infection. Mixed infection with genetically related viruses may facilitate virus recombination [80], allowing new virulent strains to emerge and/or re-emerge. The observed overall higher prevalence and coinfection rate of viruses in pigs that were reared under conventional farms compared to SPF pigs could be attributed to the improved husbandry practices of the SPF farm. However, in order to gain a comprehensive understanding of the impact of PAstV, EMCV, PKV, and PSaV infection on production, reproduction, and growth parameters of healthy pigs kept on SPF and conventional farms, studies with a larger number of farms and a wider geographical coverage are required.

## 4. Materials and Methods

### 4.1. Specimen Collection

Between January and November 2020, 483 fresh porcine fecal samples or rectal swabs were collected from apparently healthy pigs reared on conventional and SPF farms in Heilongjiang province, China. Farms that rear pigs free of specific pathogens, whose health status is often approved by a competent authority, managed in a specially designated swine facility with regular and strict disease surveillance and monitoring program, are classified as SPF farms. Swine farms that are not approved to be free of specific pathogens and do not have a specially designed facility and biosecurity systems to prevent the spread of pathogens are classified as conventional pig farms. From 8 SPF farms, 310 fecal samples from pigs of various ages, including piglets (*n* = 86), weaner (*n* = 121), fattening pigs (*n* = 74), and sows (*n* = 29), were collected. A total of 173 fecal samples from pigs of various groups, including piglets (*n* = 72), weaner (*n* = 40), fattening pigs (*n* = 40), and sows (*n* = 21), were collected from 14 conventional farms. Fresh fecal samples were stored at −80 °C in RNase-free containers until RNA isolation.

### 4.2. RNA Extraction and Reverse Transcription

Fecal samples were mixed with 10% (*w*/*v*) PBS (phosphate-buffered saline), centrifuged at 400× *g* for 20 min at 4 °C, after which supernatants were collected. According to the manufacturer’s instructions, total RNA was extracted from fecal supernatants using the TIANamp virus RNA Kit (Tiangen Biotech, Beijing, China).

Complementary DNA was synthesized in a 20 µL reverse transcription reaction mixture containing 2 µL of Golden MLV buffer, 1 µL Golden MLV enzyme, 1 µL random hexamer primers, 1 µL dNTP, 0.5 µL RNase inhibitor, 1 µg of total RNA, and RNase free double distilled water, and then incubated at 37 °C for 15 min and at 85 °C for 5 s (TakaRa, Dalian, China).

### 4.3. Primers, PCR Protocols, and Gel Imaging

The primers used for PCR and sequencing are indicated in Table 2. Before performing PCR reactions, the specificity and reliability of primers were checked by Primer-BLAST (https://www.ncbi.nlm.nih.gov/tools/primer-blast/index.cgi, accessed on 10 September 2020). PCR reaction was carried out as previously described [21,22,23,24] and PCR amplicons corresponding to each virus were visualized under UV light on a 1.5% agarose gel. Before sample screening, the specificity of primers was checked by sequencing randomly selected positive samples.

### 4.4. DNA Sequences and Phylogenetic Analysis

Based on farm type and band intensity, 10 to 20 positive samples for PAstV, EMCV, PKV, and PSaV were chosen for sequencing. PCR products were separated using agarose gel electrophoresis and then extracted using the QIAquick gel extraction kit (Qiagen, Hilden, Germany). Purified PCR products were either ligated into pMD18T vector or sequenced directly, and the nucleotide sequences were then compared to the available sequences in the NCBI nucleotide database using BLASTN (http://www.ncbi.nlm.nih.gov/BLAST, accessed on 20 January 2021). ClustalW, which is part of MEGA 7.0.26, was used to align nucleotides [81]. Phylogenetic trees were built using the neighboring method and branch statistics were calculated by bootstrap analysis of 1000 replicates. The Kimura 2-parameter model implemented in MEGA 7.0.26 was used to compute nucleotide identities [81].

### 4.5. Statistical Analysis

Data analysis was performed using Pearson’s chi-squared (*X*^2^) test with confidence limits of 95% (*p >* 0.05), using R software version 4.1.2. Figures were created using R software packages, such as ggVennDiagram, ggbeeswarm, and ggplot2.

## 5. Conclusions

In this study, PAstV, EMCV, PKV, and PSaV were found to be prevalent in pigs. The prevalence of each of these viruses depends on the farm type, age, and coinfection status. The current study demonstrates the coexistence of swine enteric viruses in healthy pigs raised on SPF and conventional farms. The detection of PAstV, EMCV, PKV, and PSaV in healthy pigs may represent a continuous yet unnoticed infection source to pigs, and possibly to humans. Hence, further studies are required in order to further understand the role of these viruses in pigs and their potential public health implications.

## Figures and Tables

**Figure 1 pathogens-11-00417-f001:**
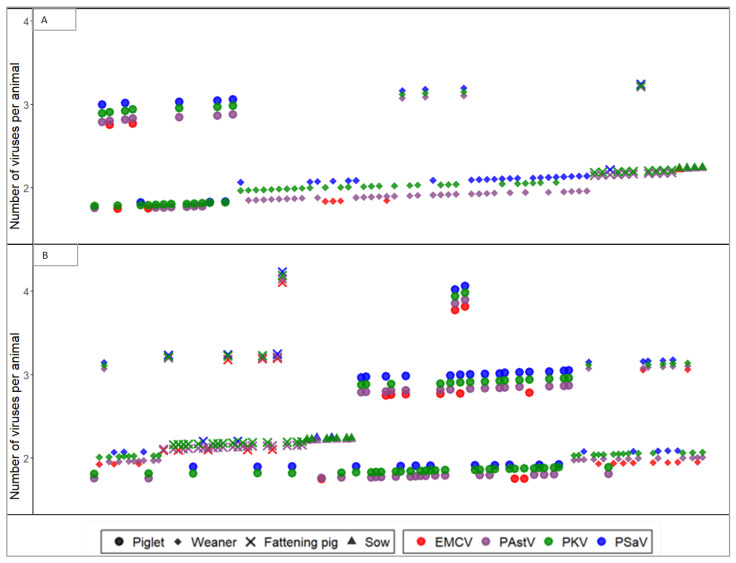
Coinfection of PAstV, EMCV, PKV, and PSaV in pigs of different age groups and farm management conditions. The coinfection status of pigs kept under SPF (**A**) and conventional (**B**) farms are visualized across the y axis. Each point represents a single virus, whereas points with the same shape aligned vertically at a specified y coordinate represent a single animal.

**Figure 2 pathogens-11-00417-f002:**
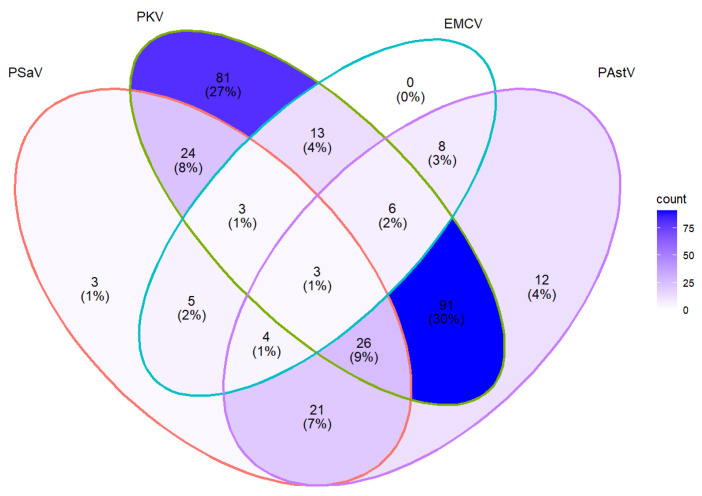
Coinfection rate of PAstV, EMCV, PKV and PSaV.

**Figure 3 pathogens-11-00417-f003:**
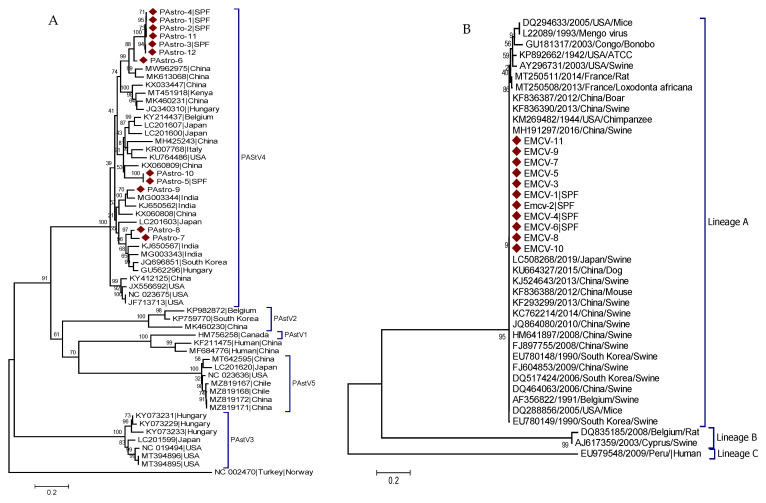
Phylogenetic analysis of PAstV, EMCV, PKV, and PSaV. The GenBank accession numbers, host, and country of origin of the reference sequence are indicated at the leaves of each phylogenetic tree. Strains detected in this study are marked with (◆), and samples from SPF farms are marked with ‘SPF’ in the leaf (tip). Scales indicate units of the number of base substitutions per site. (**A**) phylogenetic tree of PAstV, (**B**) phylogenetic tree of EMCV, (**C**) phylogenetic tree of PKV, and (**D**) phylogenetic tree of PSaV.

**Table 1 pathogens-11-00417-t001:** Summary of the prevalence of PAstV, EMCV, PKV, and PSaV.

Variable	Number of Pigs (*n*)	% (Positive/*n*)				Coinfection Rate
		PAstV	EMCV	PKV	PSaV	
Conventional farms	173	56.1 (97/173)	19.1 (33/173)	72.8 (126/173)	29.5 (51/173)	83.2 (124/149)
SPF Farms	310	23.9 (74/310)	2.9 (9/310)	39 (121/310)	12 (38/310)	53.0 (80/151)
Piglet	158	32.9 (52/158)	9.5 (15/158)	71.5 (113/158)	24.1 (38/158)	61.7 (71/115)
Weaner	161	45.3 (73/161)	10.6 (17/161)	38.5 (62/161)	25.5 (41/161)	84.8 (84/99)
Fattening pig	114	31.6 (36/114)	7.9 (9/114)	45.6 (52/114)	7 (8/114)	59.7 (37/62)
Sow	50	20 (10/50)	2 (1/50)	40 (20/50)	4 (2/50)	50.0 (12/24)
Overall prevalence		35 (171/483)	8.7 (42/483)	51 (247/483)	18 (89/483)	68.0 (204/300)

**Table 2 pathogens-11-00417-t002:** List of primers used for RT-PCR and sequencing.

Virus	Primer Name	Primer Sequence (5′ → 3′)	Target Gene	Amplicon Size	Ref.
PAstV	PAstV-F	TGACATTTTGTGGATTTACAGTT	ORF1b/ORF2	799	[21]
	PAstV-R	CACCCAGGGCTGACCA			
EMCV	ECMV F	GGTGAGAGCAAGCCTCGCAAAGACAG	3D polymerase gene	286	[22]
	ECMV R	CCCTACCTCACGGAATGG GGCAAAG			
PKV	PKoV F	TGGATTACAAGTGTTTTGATGC	3D Polyprotein gene	351	[23]
	PKoV R	TGTCGTAGAACTCCTTGATGAA			
PSaV	PSaV F	CTCATCAACCCTTTTGAAAC	Capsid protein	757	[24]
	PSaV R	AAAGCATGATGTTGTTAGGC			

## Data Availability

Data is contained within the article or Appendix A.

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
