# Peer review of "Molecular Detection and Genetic Characterization of Potential Zoonotic Swine Enteric Viruses in Northern China"

_pathogens, 2022, doi:10.3390/pathogens11040417_

Round 1
Reviewer 1 Report
Authors have detected the presence of four swine enteric viruses by RT-PCR in 483 samples from healthy pigs. Samples were collected from SPF and conventional farm. Phylogenetic analyses have been carried out for the 4 viruses.
The scope of the study is local, as samples were obtained from a Chinese province. This must be reflected in the title. Besides, the specimen collection is not clear (and this is an important point). The 310 samples were collected from an SPF farm, but the number or conventional farms is not indicated. If samples were collected from January to November, any animal was tested twice? Why were the sequenced samples randomly selected? The number of farms is very limited to make an extrapolation and to achieve a conclusion.
Results
2.1: the first two sentences (lines 79-82) are not part of the results, but of the introduction.
The third sentence is not well written. On the other hand, table 2 is the first of the tables referenced.
Fig 1 has no sense. Arbitrary number of samples for each virus are presented. The expected size is not indicated. Marker bands are not properly distinguished. It is not necessary to present a figure with an image of an electrophoresis gel.
The last sentence (lines 104-107) is not a result, but part of the discussion.
2.2
Line 112: Figure 1 is not correct; the authors refer to figure 2. Line 128 same
Figure 3 is impossible to visualize, the symbols are superimposed. this figure is not the best way to present the coinfection results. For example, it makes no sense to indicate 2.5 or 3.5 number of viruses per animal.
2.4 In the phylogenetic analysis, the origin of the samples (at least SPF vs conventional) is not indicated. Why authors use the p-distance as substitution model? It is hardly used since it underestimates the number of nucleotide changes
Discussion:
Line 290: Authors indicate that mixed infection can increase virus recombination. For EMCV and PKV can be assumed, as both are picornavirus, but can provide any reference of recombination between calicivirus and/or astrovirus and/or picornavirus?
Line 311: the suggestion of a wider virus host range an the possibility of cross-species transmission is a bit risky, as the authors only have analysed 286 nt (386 from the amplicon minus 51 of the primers)
Lines 327: A possible recombination between PSaVs found with another genogroup can not be inferred with the results obtained in this work. A recombination analysis should be done, although the sequenced region is small (717 nt).
Conclusions:
The conclusion that improving farm management practices reduces the infection rates of enteric viruses is obvious, it is expected that specific pathogens free pigs have less infections by other viruses.
Reviewer 2 Report
In the submitted manuscript " Molecular detection and genetic characterization of potential zoonotic swine enteric viruses", Gebremeskel Mamu Werid et al. analyzed the presence of 4 enteric viruses in asymptomatic pigs. The results of the paper look interesting and worth of publication after major revisions. A very deep correction of the English must be done by a native speaker.
Specific comments
INTRODUCTION
Lines 34-37 Complete the sentence, like it is now look incomplete and sounds not really scientific
Line 37: Can they be really considered a group?
Lines 40-41. “suspected of zoonosis” is a very strong claim and must be very well argumented.
RESULTS
Results often look redundant, with the same data repeated many times. Please revise.
77-78: The title looks incorrect, please revise
Lines 79-82 the sentence should be moved in the discussion section
82-83 revise
Lines 105-107 move to discussions
Figure 1 . Is there the positive control lane? If yes please clearly indicate it
Results ,Table 1 and Figure 2 Results referred to the age groups must be considered also with respect to the belonging farm and consequently described separated. Found differences could be also addressed to the kind of farm.
Lines 111-116 prevalence of EMCV and PSaV is not higher in piglets but in weaner.
Lines 118-120 It not clear the meaning, rewrite please
125-128 rewrite
Figure 2 presents data already described in the test and in the table. The same data are presented in 3 ways and it is too much.
135 the word “only” is incorrect
135-138 these percentages should be referred to positive samples instead of total samples.
136-137 Eliminate
140-141 redundant results
Figure 3. What does it mean? It is not clear and again the data look the same of those already presented in the other figures.
Line 163. What do you mean for randomly selected? Better explain. How many samples were sequenced? I think that samples to be sequenced where at least selected o the basis of band intensity or as a percentage of the total. Where samples chosen from different age groups? From different farms? Please explain.
Line 177: 61.7% could not be defined as closely related
Lines 180-183 rewrite for a better meaning
DISCUSSION
Discussion section is redundant. It must be rewritten discussing all the disease together especially because the differences described are always and only related to farm management.
Line 213. Why zoonotic? Please explain
Line 214 specify in which species
Line 239. Which other studies. Please specify
Lines 254-259 avoid to add this list of data and percentages without discussing them.
Lines 285-286 The sentence should be eliminated or correctly explained with appropriate literature.
Lines 309-312 Unclear meaning
Line 317 substitute the word indicating with suggesting
Lines 325-325 Revise and rewrite please
MATERIALS AND METHODS
Insert a paragraph explaining the main differences in management between the two kind of farms.
Lines 336-339 Indicate the number of animals belonging to different age groups for each kind of farm.
Line 346 eliminate “of”
Line 347 eliminate “and”
Paragraph 4.3
Was 1% gel enough for small amplicons?
Eliminate lines 356-359 since explained also in 4.3 paragraph.
Conclusions
Lines 382-384 Meaning not clear. Please revise
Round 2
Reviewer 1 Report
This version of the manuscript has been notably improved. Nevertheless, extensively editing of english language must be carried out. As example, the phrase “In another study, rodent outbreaks have been linked to EMCV infections in pigs” (line 276) . Rodents are not responsible, but the infection in rodents. On the other hand, I strongly recommend for future submissions that the modifications be included in the response letter, and not just indicated with the line numbers.
Reviewer 2 Report
The authors have improved the paper according to my comments. I only suggest to revise the English by a native speaker. Furthermore I still have some doubts about figure 1 that for me remains unclear. A better explanation in the legend is necessary
